# The Effectiveness of Tonsillectomy in the Diagnostic Workup of Squamous Cell Carcinoma Unknown Primary in the Head and Neck Based on p16 Immunohistochemistry

**DOI:** 10.3390/medicina60121932

**Published:** 2024-11-24

**Authors:** Patrik Stefanicka, Katarina Krupkova, Gabriela Pavlovcinová

**Affiliations:** 1Department of Otorhinolaryngology-Head and Neck Surgery, Medical Faculty, Comenius University Bratislava, 851 07 Bratislava, Slovakia; 2Department of Otorhinolaryngology-Head and Neck Surgery, Bory Hospital, 841 03 Bratislava, Slovakia; 3Department of Otorhinolaryngology, University Hospital Nitra, 950 01 Nitra, Slovakia

**Keywords:** carcinoma, squamous cell, Human Papillomavirus Viruses, oropharyngeal neoplasms, carcinoma unknown primary

## Abstract

*Background and Objectives*: Despite the distinct entity of both p16-positive and p16-negative squamous cell carcinoma unknown primary in the head and neck (HNSCCUP), the diagnostic workup did not differ. The aim of the study was to determine the effectiveness of palatine tonsillectomy in the identification of primary tumours in two groups of p16-positive and p16-negative HNSCCUP. *Materials and Methods*: Patients with HNSCCUP managed in two tertiary care referral centres from 1 January 2014 to 31 December 2020 were analysed retrospectively. *Results*: Sixty-six patients with HNSCCUP diagnosis were included consecutively. HPV status of metastatic cervical lymph nodes using immunohistochemistry with p16 protein was available for all patients. The proportion of both p16-positive and p16-negative groups was not significantly different (*p* = 0.242). Of the 39 patients who underwent palatine tonsillectomy, tonsillar cancers were revealed histologically in 6 (15.4%) patients, and all these patients were p16-positive. No primary tonsillar tumour was found in the p16-negative group (0 of 17). The primary tumour identification rate in p16-positive HNSCCUP patients using palatine tonsillectomy was 27% (6 of 22). *Conclusions*: The diagnostic workup of HNSCCUP should be guided according to HPV/p16 status. Palatine tonsillectomy is a useful procedure in identifying primary cancer in p16-positive SCCUP patients, however, its effectiveness in p16-negative patients is debatable.

## 1. Introduction

A patient with proven squamous cell carcinoma in the cervical nodes without an identified primary cancer represents an unknown primary cancer. The majority of patients with neck metastasis have an identified primary tumour after meticulous physical examination, including palpation of the base of the tongue and inspection of the skin, flexible endoscopy (with optical enhancement technologies), and after imaging. Accordingly, patients without primary tumour detection should have undergone endoscopic evaluation under general anaesthesia with at least direct biopsies of tongue base and palatine tonsillectomy being conducted, and only with such evaluation can patients be categorized as having squamous cell carcinoma unknown primary (SCCUP) [1,2].

Performing random biopsies of the nasopharynx, oropharynx, and hypopharynx in the absence of abnormal endoscopic findings is no longer recommended. This is because the yield of these biopsies is very low in contrast to directed biopsies based on endoscopic findings [2].

Palatine tonsillectomy (unilateral and bilateral) was proven to increase the identification of primary tumours and has already become the standard in the detection of primary tumour in patients with SCCUP [3,4,5,6,7,8,9].

On the other hand, lingual tonsillectomy was not a common procedure for a similar purpose because there is not favourable transoral access, and a distinct plane between the lingual tonsils and the surrounding tissue does not exist. With improving visualization of the tongue base, especially with surgical robots and with special endoscopes, transoral robotic surgery (TORS) and transoral laser microsurgery (TLM) are more frequently applied in patients with SCCUP [1,10,11,12]. Furthermore, lingual tonsillectomy, using approaches such as TORS/TLM, may improve the identification of occult primary tumours with quite low perioperative complications [13].

In systematic reviews, palatine and lingual tonsillectomies identify primary tumours in patients with SCCUP who have completely negative office examination and imaging results in approximately 32–49% and 45–56% of cases, respectively [13,14,15].

If fine needle aspiration cytology (FNAC), an important initial diagnostic procedure in the investigation of neck mass, fails to produce a diagnosis, core biopsy is a good option to confirm cancer. Achieving an accurate histologic diagnosis from the cervical mass using core cut biopsy more easily enables justifying PET/CT and panendoscopy in the operating room. It is especially important to exclude other histological diagnoses than squamous cell carcinoma (SCC) prior to performing operative endoscopy in the operating room. Excisional biopsy of a mass without completion of neck dissection offers no therapeutic advantages over a less-invasive positive core biopsy, as the patient needs the same dose of radiation. The surgeon should be prepared to perform a neck dissection at the time of open neck biopsy if the frozen section confirms SCC [1,2].

Head and neck squamous cell carcinoma unknown primary (HNSCCUP) occurs in 1 to 7% of new head and neck cancer cases [1,16,17]. Recent papers have noted that the incidence of patients with SCCUP is increasing with the increasing numbers of HPV- related oropharyngeal cancers. This increasing incidence is supported with the observation that most SCCUP are HPV-positive [18].

In patients with histologically proven SCC of the neck, it is incumbent to search for the primary site, which is usually located in the palatine or lingual tonsil, because HPV-positive oropharyngeal cancers are commonly found in the cryptal epithelium of these tonsils [2,18,19]. Identification of a primary site tumour allows us to direct appropriate treatments more specifically [1]. Detection of the primary site by applying transoral surgical approaches (TORS, TLM) will either permit a complete resection of the primary tumour or permit focused radiation, thus sparing adjacent sites in the oropharynx [1,2,13,18,20,21,22].

HPV-positive SCCUP differs from HPV-negative SCCUP in their clinic-demographic characteristics [18]. Currently, there is no clearly defined standard diagnostic algorithm for unknown primary tumours, especially for HPV-positive and HPV-negative patients [13].

The aim of the study was to determine the effectiveness of palatine tonsillectomy in the identification of primary tumours in two groups of p16-positive and p16 negative head and neck SCCUP patients.

## 2. Materials and Methods

Patients with histologically confirmed metastatic squamous cell carcinoma in the regional cervical lymph nodes and no evidence of primary tumour managed in two tertiary care referral centres in Slovakia from 1 January 2014 to 31 December 2020 were analysed retrospectively.

Inclusion criteria were the absence of a primary tumour after detailed clinical examination, including flexible endoscopy, negative cross-sectional imaging methods, negative endoscopic evaluation under general anaesthesia with directed biopsies of the tongue base, and undergoing palatine tonsillectomy. Patients with histologic diagnosis of non-squamous cell malignancy with a previous history of malignancy or a history of head and neck irradiation were excluded. Patients with suspected primary tumour on physical or radiographic examination and suspected findings during direct endoscopy were excluded if these suspicious lesions were proven histologically. Lingual tonsillectomy was not a standard part of the diagnostic workup of HNSCCUP at that time.

HPV association was examined by immunohistochemistry with p16 protein. An immunohistochemical stain with p16 protein was defined as positive in the case of at least moderate to strong nuclear and cytoplasmic expression in a majority (≥70%) of tumorous cells. Patients were divided according to the p16 immunohistochemical stain into two groups: p16-positive and p16-negative.

Demographic and clinical characteristics, the identification rate of an occult primary tumour, and the size of the identified tumours of these two groups were correlated with each other.

Statistical analysis was performed using the Chi-squared test and the Student’s t test where appropriate.

Statistical tests were two-sided and *p* < 0.05 was regarded as significant.

## 3. Results

Sixty-six patients with a diagnosis of SCCUP were included consecutively. The number of SCCUP patients per year from the years 2014 to 2020 was as follows: 10, 7, 6, 7, 13, 10, and 13. Of the 66 patients, 55 (83.3%) were male and 11 (16.7%) were female. The mean age of all patients was 61 ± 8 years (range, 40–84 y), with no significant difference in age between the males and females (61 ± 8 vs. 60 ± 11 years, *p* = 0.791). HPV status of metastatic cervical lymph nodes using immunohistochemistry with p16 protein was available for all patients, and of these 31 (47%) were p16-positive and 35 (53%) were p16-negative. The proportion of both the p16-positive and p16-negative groups was not significantly different (*p* = 0.242). The mean age of the p16-positive group was 60 ± 8 years (range, 40–72 y) and of p16-negative group was 61 ± 8 years (range, 44–84 y), which was not statistically significant (*p* = 0.989). A higher proportion of the p16-positive patients were non-smokers compared to p16-negative patients, although this was not statistically significant (63% vs. 37%, respectively, *p* = 0.098).

All patients underwent contrast-enhanced computed tomography (CT) without evidence of a primary tumour. PET-CT was carried out in 36% (24 of 66) of patients. Only three patients (13%) of this subset had suspected primary lesions with non-specific FDG uptake in the oropharynx, but no one was histologically confirmed. Another three patients with histologically proven SCC according to PET-CT findings were excluded, because only patients without suspected primary tumour on physical or radiographic examination were eligible for this study.

The N status of the 31 patients in p16-positive group was as follows: 23 (74%) N1; 2 (6%) N2; 6 (19%) N3. The N status of the 35 patients in p16-negative group was as follows: 1 (3%) N1; 16 (46%) N2; 18 (51%) N3.

Ten (15%) patients from the whole series had undergone bilateral palatine tonsillectomy in the past. Of the remaining 56 patients, 39 (59%) patients, 17 (44%) unilateral and 22 (56%) bilateral, underwent palatine tonsillectomy. From the p16-positive group, 4 (13%) patients had undergone palatine tonsillectomy in the past. Twenty-two (71%) patients underwent palatine tonsillectomy as a part of the diagnostic workup, 16 (73%) patients bilaterally and 6 (27%) unilaterally. Six (17%) patients from the p16-negative group had a history of tonsillectomy in the past. Of the remaining 29 patients, 17 (49%) patients had palatine tonsillectomy, 11 (65%) unilateral and 6 (35%) bilateral.

Of the 39 patients who underwent palatine tonsillectomy, tonsillar cancers were revealed histologically in 6 (15.4%) patients, and all these patients were p16-positive. No primary tonsillar tumour was found in the p16-negative group (0 of 17). The identification rate of primary tumour in the p16-positive SCCUP patients using palatine tonsillectomy was 27% (6 of 22). No contralateral or bilateral tonsillar cancer was found.

Postoperative hemorrhage following palatine tonsillectomy occurred in 2.5% (1 of 39) of patients. This one patient required admission to the hospital and operative intervention on postoperative day 9. There were no other complications registered.

Demographic and clinical characteristic of the series are summarized in Table 1.

The mean maximal diameter of the palatine tonsillar cancer was 10.5 mm (range, 5–20 mm).

Biopsies of tonsil fossa were negative in all patients who had undergone tonsillectomy in the past.

## 4. Discussion

Unknown primary squamous cell carcinoma of the head and neck (HNSCCUP) is a clinical entity characterized by histologically proven cervical lymph node metastases of SCC in the absence of any evidence of a primary tumour despite a comprehensive diagnostic workup. Patients must undergo a thorough diagnostic workup in order to be defined as carcinoma unknown primary, although their management is still a matter of debate.

FNAC is an initial step in the assessment of occult primary tumour of the neck after history and clinical examination. Routine assessment for p16 and Epstein-Barr virus should be performed on fine needle aspiration (FNA) specimens [20]. For patients presenting with cystic or necrotic lymph nodes, initial FNA may yield non-diagnostic results if acellular or necrotic material is sampled. Repeat biopsy under ultrasound or CT guidance, focusing on viable tissue, can often provide a definitive diagnosis without requiring more invasive surgical procedures or examination under anesthesia [23].

Open biopsy is only used after failure of repeated FNAC to lead to a diagnosis, with the intent to proceed with completion neck dissection in the case of cancer [11,20].

Targeted medical history, complete head and neck examination, including flexible endoscopy, and diagnostic imaging are essential parts of the diagnostic workup for SCCUP.

Flexible endoscopy may be complemented with advanced visualization techniques, such as narrow band imaging (NBI) to facilitate detection of potential primary sites for a targeted biopsy. In the systematic review and meta-analysis by Di Maio et al. [24], the overall detection rate of NBI was 0.35 (99% CI, 0.18–0.53), which allowed localization of the primary tumour in 61 out of 169 patients that was otherwise not detected by the usual diagnostic workup. Moreover, NBI technique may also assist in defining surgical margins and achieving complete resection [25].

An initial imaging method for patients with SCCUP is contrast-enhanced CT. If the primary tumour is not suspected after detailed physical examination and CT, PET-CT is usually recommended. To increase the detection rate of the primary tumour and decrease false positive rates, PET-CT should be scheduled prior to operative endoscopy with biopsies [20,25].

However, PET/CT has two main limitations. These are the insufficient reliability of detection of tumour smaller than 10 mm in size and the degree of physiologic fluorodeoxyglucose uptake in tonsillar tissue, which can obscure primary tumours [11,13,20,22]. Given that the size of an occult primary tumour of 10 mm or less occurred in 57–65% of patients, more than half of the unknown primary tumours may be below PET-CT detection levels [13,21,26].

The mean maximal diameter of the palatine tonsillar cancer in our histological analysis was 10.5 mm, which is comparable with other studies [13,21,26]. In a study by Zengel et al. [26], the average size of the primary tumour in the palatine tonsils was 11 mm, and in the Kubik et al. [21] analysis, the mean size of the primary tumours was 9.3 mm.

The next diagnostic step is an endoscopic examination under general anaesthesia to evaluate all mucosal sites of risk of the upper aerodigestive tract with direct biopsies of suspicious areas. Random biopsies of normal-appearing mucosa have very low yield and are no longer recommended [1,2].

The successful identification of a primary tumour employing tonsillectomy (palatine or lingual) was shown in retrospective series and systematic reviews [10,11,12,13,14,15,22]. Despite this, there are still controversies in terms of indication of tonsillectomies.

The National Cancer Center Network (NCCN) recommends unilateral or bilateral palatine tonsillectomy, biopsies, or excision of the lingual tonsils during examination under anaesthesia. Lingual tonsillectomy may be considered if the palatine tonsils are negative for tumour and other biopsies are negative [2].

The NCCN guidelines do not make specific recommendation on sidedness of the tonsillectomy. A contralateral palatine tonsillectomy identifies the primary tumour in 10–23% of cases, and therefore bilateral tonsillectomies increase the detection rate. This seems reasonable given the minimal additional morbidity of bilateral tonsillectomy to unilateral tonsillectomy [5,13,27].

In our series of 66 patients with SCCUP, 10 patients underwent bilateral tonsillectomy in the past; in 39 patients, palatine tonsillectomy was performed as a part of the diagnostic workup. Of these 39 patients, 56% (22) were bilateral and 44% (17) were unilateral tonsillectomy. Twenty-two patients underwent palatine tonsillectomy for diagnosis of SCCUP in the p16-positive group (73% bilaterally) and 17 in p16-negative group (35% bilaterally).

The systematic review and meta-analysis by Di Maio et al. [15] aimed to evaluate the effectiveness of palatine tonsillectomy in patients with cervical metastasis from SCCUP origin; the authors showed an overall tonsillectomy detection rate of 0.34 (99% confidence interval 0.23–0.46). A total of 140 occult tonsillar malignancies were identified; of these, 124 (89%) were ipsilateral, 2 (1%) were contralateral, and 14 (10%) were synchronous bilateral. Given the non-negligible number of bilateral/contralateral occult tonsillar tumours, bilateral palatine tonsillectomy should be taken into consideration in the diagnostic workup of patients with head and neck SCCUP [15].

Only a few studies compare the identification rate of primary tumour in SCCUP based on HPV status.

Podeur et al. [28], in a series of 47 patients with SCCUP with negative PET-CT and endoscopy under general anaesthesia who underwent ipsi- or bilateral palatine tonsillectomy, revealed 12 (26%) tonsil cancers (10 ipsilateral, 1 contralateral, and 1 in bilateral lymphadenopathy). Among 32 patients who had a p16 analysis and tonsillectomy (2 of the 12 tonsil cancers did not have a p16 analysis), the rate of primary detection was 59% (10/17) for p16-positives and 0% (0/15) for p16-negatives (*p* < 0.001). Oropharyngeal primaries were exclusively found in p16-positive patients and never in p16-negative patients.

Similar results were found in our study. We detected total tonsillar cancer in 15.4% (6/39) patients, all of whom were p16-positive. Therefore, the primary tumour identification rate using palatine tonsillectomy in p16-positive patients was 27% (6/22) and was 0% in p16-negative patients (0 of 17).

Currently, management of the base of tongue in HNSCCUP is more frequently discussed.

A study from Mayo Clinic Arizona by Nagel et al. [11] showed that a surgical algorithm for the unknown primary that includes TLM with lingual tonsillectomy offers the greatest likelihood of successfully detecting the location of occult primary tumours. They considered TLM as an ideal technique for performing a lingual tonsillectomy. When selecting those cases in which a TLM algorithm with lingual tonsillectomy was utilized, the detection rate was 86% (31 of 36). Tumours were most commonly found in lingual (65.0%) and palatine tonsils (27.5%). Among tumours of the tongue base, 25 of 26 tumours contained sufficient tissue for HPV testing. Twenty-three of these twenty-five tumours (92%) were p16 positive. Deep biopsies of the base of tongue were successful in 13 of 34 patients (38%) undergoing biopsies. Of those patients with negative biopsies, a lingual tonsillectomy was performed in 14 cases. Eight of these fourteen patients (57%) yielded a positive tumour even after previous negative biopsies [11].

Waltonen et al. [9] have already shown, in patients with SCCUP, that palatine tonsillectomy offers a significantly higher likelihood of finding occult tonsillar tumours than deep tonsil biopsy (29.6% vs. 3.2%, *p* < 0.0002).

In the systematic review by Fu et al. [13], which included a total of 8 studies and 139 patients, TORS/TLM identified primary tumour in 80% (111/139) of patients overall, and 67% (36/54) patients with no remarkable findings following physical exam, radiologic imaging, and panendoscopy with directed biopsies. An occult primary was identified in 60 of 108 (56%) patients undergoing lingual tonsillectomy and 34 of 70 (49%) patients undergoing palatine tonsillectomy using TORS/TLM. In patients for whom the sidedness of the tonsillectomy was specified, primary tumours were found in the contralateral palatine tonsil in 15% (2 of 13), and in the contralateral lingual tonsil in 6% (3 of 49) of patients [13].

Because of the inconstant diagnostic value of SCCUP using lingual tonsillectomy, Sudoko et al. [22] performed a retrospective study to determine this diagnostic value after using strict inclusion criteria. These included the absence of suspicious findings on physical exam, flexible endoscopy, and PET-CT, as well as negative biopsies after panendoscopy and palatine tonsillectomy. The rate of unknown primary diagnoses from lingual tonsillectomy in this study decreased to 25% (4/16) [22].

More recently, Al-lami et al. [14] conducted a larger systematic review and meta-analysis of the effectiveness of transoral surgical techniques in identifying head and neck primary cancer in carcinoma unknown primary. The primary cancer was identified in 64% (567/777) of the patients. The primary identification rates were 45% and 32% in lingual (*n* = 273) and palatine tonsillectomy (*n* = 118), respectively. Irrespective of the surgical techniques used, the authors showed that lingual tonsillectomy is a useful adjunct in identifying the primary cancer in SCCUP patients.

On the other hand, Motz et al. [18] did not find a significant increase in the detection of primary tumours in SCCUP since TORS was adopted (53.8% vs. 64.3%, respectively, *p* = 0.34).

More controversial is the sidedness of lingual tonsillectomy.

The NCCN guidelines offer the option of performing a lingual tonsillectomy while not specifying whether this can be performed unilaterally or bilaterally [20]. AHNS Guidelines recommended a unilateral lingual tonsillectomy at the time of palatine tonsillectomy when no primary can be identified. Some have advocated a bilateral lingual tonsillectomy, as 6% of contralateral lingual tonsillectomy specimens were positive for carcinoma [13,20].

In the ASCO Guidelines, recommendations differ for patients with unilateral or bilateral lymphadenopathy. Patients with unilateral lymphadenopathy should undergo ipsilateral palatine tonsillectomy. If the frozen section does not reveal a primary in the palatine tonsil, ipsilateral lingual tonsillectomy may be performed [25].

In the case of bilateral lymphadenopathy, unilateral lingual tonsillectomy is advised on the side with the greater nodal burden. If the ipsilateral procedure fails to identify the primary tumour, contralateral lingual tonsillectomy may be performed. If the lingual tonsils are negative for a primary tumour on a frozen section, palatine tonsillectomy ipsilateral to the neck with greater nodal burden may be considered. Bilateral palatine tonsillectomy after bilateral lingual tonsillectomy should be avoided because of the risk of a circumferential wound and subsequent oropharyngeal stenosis [1,25].

NCCN and ASCO guidelines for the diagnosis and management of SCCUP include palatine and lingual tonsillectomy irrespective of HPV status [2,25].

In a recent systematic review and meta-analysis, Al-lami et al. [14] found that the detection rates relating to HPV status were 82% (178/216) for HPV positive and 12% (7/59) for HPV negative tumours. This low detection rate in HPV-negative SCCUP must be carefully considered when making the decision whether to perform tonsillectomy, especially lingual tonsillectomy [14].

Kubik et al. [21] have already analysed transoral robotic lingual tonsillectomy (base-of-tongue mucosectomy) in a cohort of patient with HPV-negative unknown primary carcinoma. Twenty-three patients with p16 negative unknown primary carcinoma underwent transoral robotic lingual tonsillectomy. A pathologic analysis of the base of tongue specimens showed a primary tumour in only 3 of 23 (13%) patients. The authors concluded that TORS lingual tonsillectomy may not be indicated for HPV-negative SCCUP. For p16 negative patients, TORS may simply be generating the unnecessary risk of bleeding, pain, and cost, and potentially delaying definitive therapy [21].

Motz et al. [18] analysed 84 patients with SCCUP. Of the 75 patients with HPV tumour status available, 90.8% (68) were HPV-positive. Most patients with identified primary tumours were HPV-positive (44 of 46 (95.7%)). All identified primary tumours were found in the oropharynx; 56.3% (27) were determined to be primary tumours of the base of tongue, 41.7% (20) of the palatine tonsil, and 2.1% (1) overlapping the base of tongue and palatine tonsil [18].

In our group of 66 patients, immunohistochemistry with p16 protein from all patients was available. In contrast to previous studies with higher percentages of HPV-positive SCCUP patients, the proportion of p16-positive and p16-negative patients in our group was not significantly different (47% vs. 53%; *p* = 0.242). Still, we did not find any primary tumour in palatine tonsils in the p16-negative group of patients. Accordingly, with the HPV-negative SCCUP patients, the benefits vs. the morbidity risk in the diagnostic workup must be carefully considered.

The exhaustive investigation carried out in the SCCUP diagnostic workup should be justified by the benefit of primary tumour identification [20,25,29].

There are some important limitations of the study that must be mentioned. This is a retrospective study with inherent biases such as institutional factors and the availability of information from a clinical database of patients that may influence the results.

Though p16 immunohistochemistry is a recommended surrogate marker for HPV tumour status, a small proportion of oropharyngeal tumours are not etiologically driven by HPV yet overexpress p16 [30]. As such, confirmatory high risk HPV testing (in situ hybridization, polymerase chain reaction) might be necessary to enhance diagnostic accuracy.

Although lingual tonsillectomy has recently been recommended for the diagnosis and comprehensive management of SCCUP, it was not routinely performed during our study and thus direct institutional comparison between the effectiveness of palatine and lingual tonsillectomy is missing.

Future studies with larger sample sizes or prospective designs may be necessary to validate these results and to better understand the clinical utility of distinguishing SCCUP patients by HPV status in routine diagnostic workflows.

## 5. Conclusions

Similar diagnostic workups in the identification of primary tumour in SCCUP have recently been recommended for both p16-positive and p16-negative groups, despite being totally distinct head and neck cancer entities. Because the detection of the primary tumour in patients with SCCUP has been documented with high clinical importance, a comprehensive diagnostic workup, focused especially on the oropharynx, including palatine and lingual tonsillectomy, seems justified for p16-positive patients. However, the same comprehensive diagnostic workup, with its associated morbidity, may not be necessary for p16-negative patients, in whom the primary tumour identification rate using tonsillectomy (palatine or lingual) seems very low.

## Figures and Tables

**Table 1 medicina-60-01932-t001:** Demographic and clinical characteristics of patients.

	p16-Positive N (%)	p16-Negative N (%)	*p* Value
No. of patients	31 (47%)	35 (53%)	*p* = 0.242
Mean age ± SD	60 ± 8 y	61 ± 8 y	*p* = 0.989
Sex			
male	24 (44%)	31 (56%)	
female	7 (64%)	4 (36%)	
Tobacco consumption			
Yes	19 (40%)	28 (60%)	
No	12 (63%)	77 (37%)	*p* = 0.098
Palatine tonsillectomy			
in the past	4/31 (13%)	6/35 (17%)	
diagnostic	22/31 (71%)	17/35 (49%)	
bilateral	16/22 (73%)	6/17 (35%)	
unilateral	6/22 (27%)	11/17 (65%)	
Identification of tonsillar cancer	6/22 (27%)	0/17 (0%)	
N status			
N1	23 (74%)	1 (3%)	
N2	2 (6%)	16 (46%)	
N3	6 (19%)	18 (51%)	

## Data Availability

No new data were created.

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
