# Peer review of "The Effectiveness of Tonsillectomy in the Diagnostic Workup of Squamous Cell Carcinoma Unknown Primary in the Head and Neck Based on p16 Immunohistochemistry"

_medicina, 2024, doi:10.3390/medicina60121932_

Round 1

Reviewer 1 Report

Comments and Suggestions for Authors

The study investigates SCCUP in a cohort of 66 patients, hypothesizing that p16-positive patients may benefit from palatine tonsillectomy for primary tumor detection, while p16-negative patients derive limited benefit, potentially facing unnecessary comorbidities from this procedure. Here are several major concerns regarding methodology, sample size, and clinical relevance that need attention:

1. Although SCCUP is a relatively rare condition, the sample size of 66 patients, with primary sites identified in only 6 out of 22 p16-positive cases and none in the 17 p16-negative cases, limits the strength of the conclusions. This low identification rate does not strongly support the hypothesis, as statistical power may be insufficient to generalize findings on the efficacy of tonsillectomy in p16-differentiated SCCUP.

2. - Immunohistochemistry (IHC) of p16 protein expression is commonly used as a surrogate for HPV-related SCC, with standards by AJCC, CAP, and ASCO defining p16 overexpression as ≥70% tumor staining at moderate or higher intensity. The current methods section mentions a threshold of 75%, which should be corrected to 70% for alignment with clinical guidelines (see reference: [doi.org/10.1200/JCO.18.00684](https://doi.org/10.1200/JCO.18.00684)).  

   - Additionally, while p16 IHC is a valuable tool, a small discordance exists, with around 5% of oropharyngeal SCC cases being HPV DNA-positive but p16-negative (see reference: [doi:10.18632/oncotarget.12335](https://doi.org/10.18632/oncotarget.12335)). Consider mentioning that confirmatory high-risk HPV testing might be necessary in some cases to enhance diagnostic accuracy.

3.   FDG-PET imaging is often useful in cases of unknown primary tumors, with studies indicating that primary sites can be located in up to 30% of such cases. The study would benefit from providing information on the uptake and activity levels observed in this population to clarify how PET imaging was utilized and its role in guiding diagnostic decisions. This would add valuable data to the results section, enhancing understanding of the preoperative diagnostic workflow.

4. Currently, the results section is relatively brief (approximately one page), whereas the discussion extends over four pages. Consider transferring findings that are described in the discussion to the results to streamline content and provide clearer separation between findings and interpretative commentary.

5.  For patients presenting with cystic or necrotic lymph nodes, initial fine-needle aspiration (FNA) may yield non-diagnostic results if necrotic material is sampled. Repeat biopsy under ultrasound or CT guidance, focusing on viable tissue, can often provide a definitive diagnosis without requiring more invasive surgical procedures or examination under anesthesia. The review could benefit from integrating these diagnostic considerations into the methods or discussion sections.

6.  Epstein-Barr virus (EBV) status of patients was not mentioned, which may be relevant given its association with certain head and neck cancers. Please include EBV status information in the results, or explain its absence if not assessed.

 Furthermore, the study does not report lingual tonsillectomy results, despite ASCO and other guidelines recommending it for comprehensive SCCUP workup. If lingual tonsillectomy was not performed, a clear rationale should be provided, as this omission may impact the study's generalizability and adherence to clinical guidelines.

7. In the section, "In our series of 66 patients with SCCUP...,” the report mentions that 29 patients in the p16-negative group underwent tonsillectomy. Please clarify whether this number is correct, as it appears inconsistent with the previously stated figures.

Similarly, the reference to Podeur et al. requires clarification. They detected 12 cases of tonsillar cancer, with 10 being HPV-positive; however, it is implied that the remaining two cases may not have undergone HPV testing. Clearer specification of these patients' testing status would enhance understanding of HPV’s relevance in SCCUP diagnosis.

8. Lastly, while this study contributes valuable insights into SCCUP diagnostic practices, the low incidence of primary tumor detection in this cohort, particularly among p16-negative cases, raises questions about the generalizability of the findings. Future studies with larger sample sizes or prospective designs may be necessary to validate these results and to better understand the clinical utility of distinguishing SCCUP patients by HPV status in routine diagnostic workflows. Please mention all the limitations of the study.

Reviewer 2 Report

Comments and Suggestions for Authors

The study examines the efficacy of palatine tonsillectomy in identifying primary tumours in two patient groups of p16-positive and p16-negative HNSCC. The topic of the study is interesting, and the article may have its scientific relevance after an adequate review.

Review:

1)      The conclusions in the abstract should be reworded. Stating that it is the p16 immunohistochemistry status on cervical lymph nodes that guides the diagnostic workup of these tumours and emphasizing the role of palatine tonsillectomy in the p16 positive group.

2)      Add to the results and tables the N (TNM) status of the patients at the time of starting the diagnostic workup.

3)      Please provide more details about the diagnostic workup of the patients, what type of imaging they had, how many had CT, MRI and/or PET-CT.

4)      Make Table 1 more uniform by putting the percentages in brackets.

5)      Write down whether there have been any complications following palatine tonsillectomy in patients who have had it.

6)      The discussion is written in a crude manner and in a non-academic English. It uses up-to-date references. For example, for NBI there is a meta-analysis that evaluates its role in head and neck SCCUP (Narrow band imaging in head and neck unknown primary carcinoma: A systematic review and meta-analysis).

7)      When you talk about a study or a systematic review always cite the authors in the text. For example, in the systematic review by ... et. al. ...

8)      Compare and contrast your results with those in other literature studies in a clear and concise manner.

Comments on the Quality of English Language

The quality of English used is very low and not very academic.

Round 2

Reviewer 1 Report

Comments and Suggestions for Authors

The authors have addressed all my concerns

Reviewer 2 Report

Comments and Suggestions for Authors

The manuscript has been appropriately edited.